# Key Roles of De-Domestication and Novel Mutation in Origin and Diversification of Global Weedy Rice

**DOI:** 10.3390/biology10090828

**Published:** 2021-08-26

**Authors:** Yong-Qing Zhu, Jia Fang, Ying Wang, Li-Hao Pang, Bao-Rong Lu

**Affiliations:** Ministry of Education Key Laboratory for Biodiversity Science and Ecological Engineering, Department of Ecology and Evolutionary Biology, Fudan University, Songhu Road 2005, Shanghai 200438, China; yq_zhu@fudan.edu.cn (Y.-Q.Z.); fangjia@fudan.edu.cn (J.F.); wangying09@fudan.edu.cn (Y.W.); 20110700002@fudan.edu.cn (L.-H.P.)

**Keywords:** differentiation, feral origin, genetic structure, introgression, networking analysis, seed shattering, weed evolution

## Abstract

**Simple Summary:**

Weedy rice is a noxious weed infesting rice fields worldwide and causing tremendous losses of rice yield and quality. The control of this conspecific weed is difficult owing to abundant genetic diversity associated with its complex origins and evolution. Applying different molecular methods, we demonstrate the multiple origins of weedy rice with the major pathway from its cultivar progenitors. The origin and diversification of weedy rice are also closely associated with differentiation of *indica*-*japonica* rice varieties. In addition, novel mutations are identified, which may promote continued evolution and genetic diversity of weedy rice. Knowledge generated from this study provides deep insights into the origin and evolution of conspecific weeds, in addition to the design of effective measures to control these weeds.

**Abstract:**

Agricultural weeds pose great challenges to sustainable crop production, owing to their complex origins and abundant genetic diversity. Weedy rice (WD) infests rice fields worldwide causing tremendous losses of rice yield/quality. To explore WD origins and evolution, we analyzed DNA sequence polymorphisms of the seed shattering genes (*sh4* and *qsh1*) in weedy, wild, and cultivated rice from a worldwide distribution. We also used microsatellite and insertion/deletion molecular fingerprinting to determine their genetic relationship and structure. Results indicate multiple origins of WD with most samples having evolved from their cultivated progenitors and a few samples from wild rice. WD that evolved from de-domestication showed distinct genetic structures associated with *indica* and *japonica* rice differentiation. In addition, the weed-unique haplotypes that were only identified in the WD samples suggest their novel mutations. Findings in this study demonstrate the key role of de-domestication in WD origins, in which *indica* and *japonica* cultivars stimulated further evolution and divergence of WD in various agroecosystems. Furthermore, novel mutations promote continued evolution and genetic diversity of WD adapting to different environments. Knowledge generated from this study provides deep insights into the origin and evolution of conspecific weeds, in addition to the design of effective measures to control these weeds.

## 1. Introduction

Agricultural weeds post great threats to global food security because of their extensive and strong competition with cooccurring crops for resources such as water, nutrients, and solar energy [1,2,3]. In addition, many weeds may also host diseases and attract insects, which likely causes outbreaks of different diseases and insect pests under certain environmental conditions [4,5]. Consequently, the widespread infestation of various weeds in agricultural fields can cause substantial yield losses of cultivated crops [6,7]. Statistical analyses by the Food and Agricultural Organization (FAO) of the United Nations indicated that yield losses caused by the infestation of weeds are much greater than the total losses of crop yields caused by diseases and insect pests [8]. Therefore, the effective control and management of agricultural weeds is always challenging, which hampers the high-output and sustainable production of various crops in agroecosystems.

Usually, weeds encompass abundant genetic diversity and wide adaptation, which is more likely associated with their complex origins and evolutionary processes [9,10,11], although such genetic diversity and adaptation harbored by weeds can also be utilized in crop breeding as valuable germplasm [12,13]. This may be particularly true for conspecific weeds that belong to the same species of their crops biologically [13,14]. Conspecific weeds and their crop/wild progenitors have a close genetic relationship, providing opportunities for frequent weed-crop/wild gene flow [15], which enhances genetic diversification and evolution of the conspecific weeds in agroecosystems through receiving novel traits (genes) from both their wild and crop progenitors [15,16]. In addition, crop volunteers with developed weedy characteristics such as seed shattering and dormancy can also evolve into conspecific weeds through the so-called de-domestication process [11,17,18]. However, our knowledge of the de-domestication process is still very limited.

Weedy rice (*Oryza sativa* f. *spontanea*, also referred to as red rice) is a conspecific weed infesting rice (*O. sativa*) fields worldwide [1,6,19]. Commonly, the infestation of weedy rice causes significant yield and quality losses of cultivated rice, resulting in significant reduction of farmers’ incomes [6]. Unlike its cultivated progenitor, weedy rice has strong seed shattering and seed dormancy [6,20,21], which enables the long-term persistence of its weeds in soil seed-banks. This weed has great morphological and genetic variation, mimicking its wild and cultivated progenitors. Altogether, these characteristics make the control and management of weedy rice extremely difficult [21,22]. It is known that weedy rice evolves from two pathways. The first pathway involves the wild *Oryza* progenitor [17,19,21], which is defined as the wild-introgression origin in this study. The second pathway involves only the cultivated rice progenitor [17,19,23,24], and is defined as the de-domestication origin. However, our knowledge concerning which is the major pathway for weedy rice origins and how weedy rice further evolves after its origins in different agroecosystems is still insufficient.

Seed shattering is a key trait associated with the origin of weedy rice, apart from its link to the origin of cultivated rice. The loss of seed shattering is the critical step for rice domestication in early stages [25,26], whereas the regaining of seed shattering promotes the reverse of cultivated rice to weedy rice through de-domestication [18,27]. To date, several seed-shattering loci, including *sh4* and *qsh1*, have been cloned [26,28,29]. The wild type of *sh4* is a transcription factor encoding the Myb3 DNA-binging domain that is involved in the establishment of an abscission layer at the base of a spikelet, releasing rice grains from panicles [26]. A mutant containing a single nucleotide substitution (G to T) of *sh4* resulted in incomplete development of the abscission layer, converting rice plants from seed shattering to seed persistent phenotypes [26]. The *qsh1* gene encodes a BELL-type homeobox that is also associated with the abscission layer formation. A single nucleotide mutation in the 5′ upstream regulatory region of the *qsh1* gene caused the loss of seed shattering [28].

Previous studies indicated the occurrences of population genetic bottlenecks during the process of rice domestication [30]. The genetic bottlenecks can significantly reduce single nucleotide polymorphisms (SNPs) at a DNA fragment of the candidate genes, such as the seed-shattering genes in cultivated rice, compared to SNPs at the DNA fragment of its wild progenitor [31]. We therefore hypothesize that weedy rice populations with the wild-introgression origin should have a comparable level of SNPs as their wild progenitor. In contrast, weedy rice populations with a de-domestication origin have a limited level of SNPs compared to their cultivated progenitor (see Figure 1). Thus, analyzing the SNPs at the seed-shattering loci and their flanking regions will provide insights for understanding the wild-introgression and de-domestication origins of weedy rice populations.

In this study, we analyzed SNPs of the two seed shattering genes (*sh4* and *qsh1*) of weedy rice populations, in comparison with those of their wild and cultivated progenitors from a worldwide geographic distribution. In addition, we also used microsatellite (simple sequence repeat, SSR) and InDel (insertion/deletion) molecular fingerprints to examine the genetic relationship and structure of the weedy, wild, and cultivated rice samples included in this study. The primary objectives of this study are to address the following questions: (1) Is wild-introgression or de-domestication the major pathway for the origin of worldwide weedy rice? (2) Is weedy rice with de-domestication origins linked to *indica*-*japonica* differentiation? (3) Does weedy rice further evolve through novel mutations to adapt to divergent environment conditions after its origin? The answers to these questions have important implications not only for understanding the role of wild-introgression and de-domestication in the origins of this conspecific weed, but also for designing strategies for effective control and management of weedy rice.

## 2. Materials and Methods

### 2.1. Plant Materials

In this study, a total of 169 weedy rice (coded as WD) samples obtained from a wide range of geographic locations worldwide were included for the DNA sequencing analyses (Appendix A). In addition, a total of 85 samples of wild rice *O. rufipogon* complex (including *O. rufipogon* and *O. nivara*, coded as W) and 167 samples of cultivated rice (coded as CV), including the two subspecies *indica* and *japonica*, used as references from worldwide range, were also included for the analyses (Appendix A). One individual or sample was randomly selected to represent each W/CV population or variety. Leaf tissues from young seedlings generated from germinated seeds grown in a greenhouse were sampled for DNA extraction.

In addition, the published DNA sequences of the *sh4* and *qsh1* loci obtained from 57 weedy rice accessions, 102 cultivated rice accessions, and 126 W accessions from GenBank (https://www.ncbi.nlm.nih.gov/genbank/, accessed on 30 July 2021) were included for analyses (Appendix A).

### 2.2. PCR Amplification and Sequencing

For the *sh4* locus (Os04g0670900), the PCR amplified DNA fragment (2509-bp) included a 1509-bp 5′ upstream region, the entire fragment of the first exon (847 bp), and a partial intron (153 bp) based on Zhang et al. [31]. For the *qsh1* locus (Os01g0848400), the amplified DNA fragment included a part of the 5′ upstream region with 1558 bp surrounding the previously reported function SNP [31].

The PCR primers used to amplify DNA fragments of the two seed shattering loci and the amplification systems followed the description of Zhang et al. [31]. The internal sequencing primers were designed using Primer3Plus [32], based on rice genome sequences [33,34]. For weedy rice and cultivated rice that are predominantly self-pollinated taxa, the purified PCR products were directly sequenced in both directions. For W samples with a considerable outcrossing rate, the purified PCR products were cloned into pMD18-T vectors (Takara, Dalian, China). Eight clones from each sample were randomly selected for DNA sequencing. Singletons and ambiguous sites were re-sequenced to assure the quality of sequences used for analyses. All obtained DNA sequences in this study were deposited in GenBank (accession numbers: see Appendix A). In addition, all the published and obtained DNA sequences were aligned using the software CLUSTAL_X [35].

### 2.3. SSR and InDel Genotyping

Fifty microsatellite (SSR) primer pairs distributed across the rice genome with more than four loci on each chromosome were used for the analyses of genetic structure [36]. Thirty-four insertion/deletion (InDel) primer pairs that can effectively identify *indica* and *japonica* genotypes of CV, W, and WD populations [37,38] were used to determine the *indica*-*japonica* differentiation in weedy rice samples. All the SSR and InDel forward primers were fluorescently labeled by either FAM (blue), ROX (red), or JOE (green).

The PCR reaction systems for SSR and InDel primers contained 1× buffer (with Mg^2+^), 0.2 mM each of dNTP, 0.2 μM of SSR primers, 20 ng of genomic DNA, and 0.3 U of Taq polymerase (Takara, Dalian, China). The reaction procedure was programmed as follows: an initial denaturation period of 4 min at 94 °C, followed by 28 cycles of 30 s at 94 °C, 30 s at 50–63 °C (primer-sequence dependent), 40 s at 72 °C, and then 7 min at 72 °C for the final extension. The amplified PCR products with different sizes and contrasting fluorescent labels were multiplexed and then separated by capillary electrophoresis on an ABI 3130 DNA Analyzer (Thermo Fisher Scientific, Waltham, MA, USA).

### 2.4. Data Score and Analyses

DNA sequences of W, WD, and CV samples were edited and assembled by the software Seqman II (DNASTAR). The ends of resulted amplicons were trimmed to remove the low-quality sequences. Sequence alignments were performed using CLUSTAL_X [35]. All aligned sequences were imported into the DnaSP package [39] to extract all nucleotide polymorphic sites, based on which various haplotypes were determined. A haplotype network was constructed using the program NETWORK 4.5.1.0 using the median joining (MJ) method [40].

SSR and InDel alleles were scored using the software GENEMAPPER Version 3.7 (Applied Biosystems). Genetic structure of W, WD, and CV samples was analyzed using the software STRUCTURE Ver. 2.3.4 [41] based on the SSR and InDel genotypic data matrixes. The STRUCTURE running parameters were set as follows: an initial burn-in run of 100,000 steps followed by a run length of 1,000,000 steps and a model allowing for admixture and correlated allele frequencies between populations. The appropriate K value was determined by running the data matrix in STRUCTURE 20 times following the principle of Evanno et al. [42]. Graphical depiction of STRUCTURE results was produced using DISTRUCT software ver. 1.1 [43].

## 3. Results

### 3.1. Sequence Characterization of the sh4 and qsh1 Loci

We obtained a total of 597 DNA sequences of the *sh4* seed-shattering locus, representing W (166), WD (228), and CV (203) samples from a worldwide geographical range (Appendix A). The analyzed *sh4* sequences showed substantially greater polymorphisms in W samples than in WD and CV samples (Table 1, Appendix A). A total of 101 variable sites were detected in the W sequences, but a much lower number of variable sites was detected in the WD (35) and CV (4) sequences. Based on these variable sites, a total of 114 haplotypes were identified for the *sh4* locus, including 103 W haplotypes (Hs1~Hs103), 9 WD haplotypes (Hs1, Hs2, Hs104~HS110), and 6 CV haplotypes (Hs1, Hs2, Hs111~Hs114) (Table 1, Appendix A).

In addition, a total of 336 sequences were obtained for the *qsh1* locus, representing W (89), WD (144), and CV (103) samples from the world collections (Appendix A). Noticeably, slightly greater sequence polymorphisms were found in the W and WD samples, with 70 and 33 variable sites, respectively, than those in the CV samples, with 24 variable sites (Table 1, Appendix A). Based on these variable sites, a total of 80 haplotypes were identified for the *qsh1* locus, including 54 W haplotypes (Hq1~Hq54), 25 WD haplotypes (Hq1~Hq5, Hq8, Hq9, Hq55~Hq72), and 15 CV haplotypes (Hq1~Hq4, Hq55~Hq57, Hq73~Hq80) (Table 1, Appendix A).

The aligned DNA sequences of the two seed-shattering loci (*sh4* and *qsh1*) showed quite different distribution patterns of haplotypes in the W, WD, and CV samples. For the *sh4* locus, two key haplotypes (Hs1 and Hs2) were found to be shared by all three taxa used in this study, accounting for 18.68% by W, 85.09% by WD, and 94.55% by CV samples (grey bars in Figure 2, Appendix A). In addition, there were many other unique haplotypes that were only identified either in W or WD or CV samples. We artificially defined these unique haplotypes as W-unique (color-coded as green), WD-unique (coded as red), and CV-unique haplotypes (coded as yellow), respectively (Figure 2). Noticeably, the proportion of the W-unique haplotypes was substantially higher in wild rice samples than the WD- and CV-unique haplotypes in the WD and CV samples, respectively (Figure 2, Appendix A). The findings confirmed the hypothesis of a genetic bottleneck in the CV gene-pool (Figure 1), and the fact that the composition of haplotypes was greatly similar in the WD and CV gene-pools.

For the *qsh1* locus, four key haplotypes (Hq1~Hq4) were found to be shared by all taxa used in this study, accounting for 22.47% by W, 80.56% by WD, and 68.93% by CV samples (grey bars in Figure 3, Appendix A). Unlike the pattern revealed by the *sh4* sequences, three special haplotypes (Hq5, Hq8, Hq9) were only shared by W and WD samples, whereas other three special haplotypes (Hq55~Hq57) were only shared by WD and CV samples (Figure 3, Appendix A). The two distinct patterns of the shared haplotypes suggested that some wild and weedy rice samples, or weedy and cultivated rice samples, had close genetic or evolutionary relationships in terms of their special haplotypes. Similarly, the proportion of W-unique haplotypes (coded as green) was substantially higher in the wild rice samples than the WD- and CV-unique haplotypes in the weedy and cultivated rice samples, respectively (Figure 3, Appendix A).

Furthermore, except for being shared by wild rice samples, the two identified haplotypes (Hq1, Hq3) were shared by weedy rice and *indica* and *japonica* types of cultivated rice that represented most of the studied samples, suggesting a close relationship between the three taxa in terms of the two shared haplotypes. Interestingly, the four special haplotypes (Hq2, Hq4, Hq56, Hq57) were only shared by WD and *indica* CV (CV-I) samples, whereas one special haplotype (Hq55) that encompassed >34% of *japonica* CV (CV-J) samples was only shared by WD and CV-J samples (Figure 4, Appendix A). This finding suggested close genetic or evolutionary relationships of some weedy and cultivated rice samples, in addition to the independent origins of these *indica*-type (coded as light-brown bar, in Figure 4) and *japonica*-type (coded as light-blue bar, in Figure 4) of WD samples.

### 3.2. Haplotype Network Analyses of the sh4 and qsh1 Sequences

We conducted network analyses of the aligned sequences of the two seed shattering loci (*sh4* and *qsh1*) to reveal the possible origins of WD accessions either from W or CV. Results from the network analyses clearly indicated that the majority of the WD samples (coded as red) were associated CV samples (coded as yellow) but only a few WD samples were nested in the W samples (coded as green) (Figure 5 and Figure 6).

Based on the network analysis of the *sh4* locus, including 114 haplotypes generated from W, WD, and CV samples, two major groups were identified (Figure 5). One group on the left mainly contained the Hs1 (the largest circle) and Hs2 (the second largest circle) haplotypes in addition to other haplotypes that were represented by >80% weedy rice samples used in this study. Obviously, Hs1 and Hs2 were the most common haplotypes shared by W, WD, and CW samples. Results from this group only suggested the possible origins of WD from either W or CV with no final resolution. Another group on the right included divergent haplotypes in which only three identified WD haplotypes (Hs108, Hs109, and Hs110) were nested in the W haplotypes, strongly suggesting the origin of these WD samples from wild rice. In addition, the three WD samples (code: 104700, YW7700, 105362 in Appendix A) were collected from India, Malaysia, and Thailand with the natural distribution of wild rice, supporting their possible wild-introgression origin.

Based on the network analysis of the *qsh1* locus, including 80 haplotypes generated from W, WD, and CV samples, more complex grouping was identified (Figure 6). Three major haplotypes (Hq1, Hq2, and Hq3) shared by W, WD, and CV samples were found to dominate the *qsh1* network. Interestingly, only a few haplotypes such as Hq5, Hq8, and Hq9 were shared only by the W and WD samples, suggesting the possible origin of these WD samples from wild rice. Coincidently, WD samples with the Hq5, Hq8, and Hq9 haplotypes were collected from India, Bangladesh, and Myanmar with the natural distribution of wild rice. In contrast, other haplotypes such as Hq55, Hq56, and Hq57 were shared only by WD and CV samples, suggesting the possible origin of these WD samples from cultivated rice. Interestingly, WD samples with the Hq55, Hq56, and Hq57 haplotypes were collected from temperate regions in northeastern China, Spain, and North Korea with the natural distribution of wild rice (Appendix A). Altogether, these results evidently indicated the independent origins of WD samples from both the wild-introgression and de-domestication pathways.

However, it was not possible to resolve the wild-introgression or de-domestication origin of most weedy rice samples that shared the Hs1 and Hs2 haplotypes for the *sh4* locus or the Hq1, Hq2, and Hq3 haplotypes for the *qsh1* locus, only based on the haplotype network analyses. Further analyses including other molecular markers such as SSR and InDel should be undertaken to reveal the wild-introgression or de-domestication origins of the WD samples with these haplotypes.

### 3.3. Genetic Structure of Wild, Cultivated, and Weedy Rice Based on SSR and InDel Molecular Fingerprints

We conducted the STRUCTURE analysis including W, WD, and CV samples that shared the Hs1 and Hs2 haplotypes from the *sh4* network analyses (Figure 5) to determine the wild-introgression or de-domestication origins of weedy rice. Results based on the randomly selected 50 SSR (simple sequence repeat) loci across the rice genome indicated a clear genetic structure of the included W, CV, and WD samples (Figure 7). When the proper K value was 3, we identified the major distinct genetic component for W (represented by blue) and for CV samples (green and red), although with introgression of other components. These results suggested substantial differentiation between the W and CV samples. Evidently, some WD samples shared the identical genetic component (blue) with the W samples, whereas other WD samples shared the identical genetic components (green and red) with the CV samples (lower panel in Figure 7). Based on our calculation, ~20% of WD samples evolved from wild rice, while ~80% of WD samples evolved from CV rice. These results clearly indicate the independent origins of the WD samples with both the wild-introgression and de-domestication pathways.

An obvious admixture was found in the genetic structure of W, CV, and WD samples, indicating frequent genetic introgression between wild, weedy, and cultivated rice. In addition, two distinct genetic components (represented by green and red in Figure 7) were identified both in CV and WD groups. The two identified genetic components were likely associated with the genetic differentiation of the *indica* and *japonica* ecotypes (or subspecies) of cultivated rice, which needed further confirmation using proper molecular fingerprints.

To further determine WD samples associated with the *indica-japonica* genetic differentiation, we conducted the STRUCTURE analysis using the insertion/deletion (InDel) molecular fingerprints developed to identify *indica* and *japonica* rice genotypes. WD and CV samples that shared the Hs1 and Hs2 haplotypes from the *sh4* network analysis (Figure 5, Appendix A) were randomly selected for the STRUCTURE analysis, in addition to previously identified standard CV-I and CV-J as references. When the proper K value was 2 owing to the bimorphic feature of the InDel markers, two distinct genetic components for standard *indica* (CV-I, red) and *japonica* (CV-J, green) varieties (Figure 8) were identified.

Accordingly, the selected WD and CV samples also showed the two distinct genetic components associated with those of the standard CV-I and CV-J samples (Figure 8). We detected ~58% WD samples with the same genetic component of *indica* CV rice, and ~42% samples of *japonica* CV rice, based on our calculation. These results confirm the *indica-japonica* genetic differentiation of the de-domestication originated WD samples, which was associated with the genetic differentiation of *indica* and *japonica* CV samples and the standard CV-I and CN-J varieties. Similarly, the admixture was also identified in the genetic structure of a small number of WD and CV samples, suggesting genetic introgression between *indica* and *japonica* genotypes both in WD and CV samples.

## 4. Discussion

### 4.1. De-Domestication Plays the Major Role in the Origin and Evolution of Weedy Rice

Our results based on the detailed comparison of the observed variable sites and characterized haplotypes of the *sh4* and *qsh1* seed-shattering genes demonstrate the multiple origins of weedy rice occurring in worldwide rice agroecosystems. In other words, weedy rice evolved both from wild-introgression origins involving its wild progenitor and de-domestication origins involving only its cultivated progenitor. The conclusion is based on the evidence that some weedy rice samples shared identical variable sites and haplotypes with the wild rice progenitor, whereas other weedy rice samples shared identical variable sites and haplotypes with the cultivated rice progenitor. Noticeably, supporting our hypothesis (Figure 1), results for the wild rice samples in this study showed much higher numbers of variable sites (101) and haplotypes (103) than the cultivated rice samples that had a substantially reduced number of variable sites (4) and haplotypes (6). Interestingly, the detected number of variable sites (35) and haplotypes (9) in weedy rice samples was greater than those in cultivated rice samples, but dramatically reduced when compared with those in the wild rice samples. These findings not only confirm the multiple origins—wild-introgression and de-domestication origins of weedy rice—but also suggest a greater proportion of weedy rice samples that are likely evolved from their cultivated rice progenitor.

The question concerning whether the major pathway of weedy rice evolved through wild-introgression involving wild *Oryza* progenitors or de-domestication involving only cultivated rice on a worldwide scale is still not completely resolved. Two previous studies reported the possibility of de-domestication origins of most USA weedy rice where no wild rice progenitors are naturally distributed [18,44]. In addition, a few other studies also proposed the likelihood of de-domestication origins of weedy rice in Asian rice fields [19,45]. Therefore, clarifying the major pathway of worldwide weedy rice origins is important for better understanding the origin and evolution of agricultural weeds, particularly conspecific weeds.

The haplotype network analyses of the *sh4* and *qsh1* genes further indicated the presence of only a few weedy rice samples that were nested among the wild rice samples. Contrarily, most weedy rice samples were closely associated with cultivated rice samples, although the origins of many weedy rice samples included in Hs1 and Hs2 of the *sh4* gene, and in Hq1, Hq2, and Hq3 of the *qsh1* gene still need further analyses for the final resolution using other types of molecular fingerprints such as SSRs and InDels. Nevertheless, these findings suggest that most weedy rice populations found in worldwide rice fields were likely originated directly from their cultivated rice progenitors without the involvement of wild rice. In other words, the major pathway of the worldwide weedy rice origins is more likely through de-domestication of cultivated rice, which well addressed our first question raised in the Introduction.

Further molecular analyses of the genetic structure of weedy rice samples, which did not show clear resolution of origins from the haplotype networks, also indicated the presence of weedy rice samples with both wild-introgression and de-domestication origins based on the SSR molecular fingerprints. Importantly, only a small proportion of weedy rice samples that were collected from tropical regions such as India, Cambodia, the Philippines, Myanmar, and Bangladesh, where wild rice populations were naturally distributed, shared the key genetic component with wild rice. The major proportion of weedy rice samples share identical key genetic components and show close genetic relationships with the cultivated rice samples (Figure 7) that were collected essentially from the temperate regions where no natural distribution of wild rice was reported.

These findings support our conclusion that de-domestication plays the major role in the origins of weedy rice occurring worldwide without the involvement of wild rice. Previous studies proposed the possibility of the de-domestication of weedy rice from the temperate rice planting regions, such as northern China and southern Europe [18,19], although the authors of these studies did not discuss the significant role of de-domestication in the origins of weedy rice. Altogether, these findings will be important for us to understand the human influenced origins and evolution of weeds in agroecosystems, particularly for conspecific weeds, in addition to designing effective strategies to control and manage these weeds. The conclusion addresses our first question in the Introduction concerning the proportion of weedy rice origins, in which most weedy rice populations around the world evolved from cultivated rice through de-domestication.

Given that most worldwide weedy rice has de-domestication origins, it is necessary to avoid or reduce seed-shattering during rice harvesting and to clear the soil seed banks by ploughing and raking paddy fields properly. These practices may considerably reduce weedy rice populations. In addition, growing a mixture of genetically divergent rice varieties in a region is not encouraged because such practices may promote inter-varietal gene flow and produce widely segregated hybrid descendants that become the sources for weedy rice origin and evolution [27,46]. Another example has shown that weedy rice populations with de-domestication origins do not usually have primary seed dormancy [11]. Therefore, it is possible to control this type of weedy rice by using the so-called “inducing-germination and killing-seedling” strategy, meaning to kill weedy rice seedlings by spraying herbicides a week after rice fields are irrigated and before rice is cultivated [7].

### 4.2. De-Domesticated Weedy Rice Associated with Genetic Divergence of Rice Varieties

It is well-known that cultivated rice has differentiated into genetically diverged types and great number of varieties [37,38,47,48,49]. *I**ndica-japonica* differentiation represents one of such substantial variations [50,51]. Usually, the *i**ndica* ecotypes (or subspecies) are grown in the tropical and subtropical rice-planting regions [37,38], whereas the *japonica* ecotypes are cultivated in the temperate rice-planting regions [37]. Interestingly, our results from this study indicate that weedy rice samples with de-domestication origins are clearly divided into two distinct groups that are hypothetically associated with either *indica* or *japonica* cultivars based on the STRUCTURE analyses using SSR molecular fingerprints (Figure 7). Further analyses of these weedy rice samples including the standard *i**ndica* and *japonica* varieties as references using the InDel molecular markers (Figure 8) confirmed their association with *indica* and *japonica* varieties. The findings based on the two sets of independent molecular fingerprints (SSR and InDel) suggest that weedy rice populations with de-domestication origins from different locations evolved from either *indica* or *japonica* rice cultivars independently. Coincidently, our results from the analysis of the proportion of different *qsh1* haplotypes (Figure 4) also support the independent origins of weedy rice from either *indica* or *japonica* cultivars. Another explanation for such findings is the possibility of hybridization-introgression of *indica* or *japonica* rice cultivars with their cooccurring weedy rice populations distributed worldwide [18,37,38,44].

Weedy rice samples or populations characterized as the *indica* or *japonica* genotypes were reported in previous studies [37,38,52]. It is important to point out that the presence of *indica* or *japonica* genotypes of weedy rice is commonly associated with the types of their cooccurring cultivated rice varieties. In other words, weedy rice populations with the *indica* genotypes are most likely found in rice-planting regions where *indica* varieties are grown, and vice versa [13,37,38]. As shown in this study, weedy rice samples sharing the same genetic component as that of standard *indica* cultivated rice were basically collected from tropical rice-planting regions such as SE Asia, Sri Lanka, and southern China, where *indica* rice varieties were predominantly grown. Conversely, weedy rice samples sharing the same genetic component as that of standard *japonica* cultivated rice were essentially collected from temperate regions such as northern China, where only *japonica* rice varieties were grown. Altogether, these findings not only confirm the major role of de-domestication in weedy rice origins, but also strongly suggest the differential origins of the de-domesticated weedy rice populations that are closely linked with their cooccurring *indica* and *japonica* ecotypes of cultivated rice.

These findings address our second question raised in the Introduction about the *indica* and *japonica* association/differentiation of the weedy rice populations with de-domestication origins. Considering that the cultivated rice progenitors contain tremendous genetic diversity globally, the complex origins and evolution processes can create greatly divergent types of weedy rice through acquiring different genes of divergent cultivated rice through de-domestication and crop-to-weed gene flow in different rice ecosystems. Such processes can easily explain the independent origins of weedy rice populations that are closely associated with *indica* and *japonica* differentiation, in addition to the abundant genetic diversity of weedy rice worldwide. Therefore, de-domestication plays a very important role in the origin and evolution of weedy rice populations, although determining the underlying reasons causing genetic differentiation and diversity of worldwide weedy rice by de-domestication or/and crop-to-weed gene flow still needs further investigation.

### 4.3. Novel Mutations Promote Further Evolution of Weedy Rice after Its Origins

Our results based on the analyses of the proportion of different haplotypes further indicate that some weedy rice samples, particularly those from southern Europe and northern China, contain the WD-unique haplotypes that are not found either in wild or cultivated rice samples in the target DNA regions of the *sh4* and *qsh1* genes (Figure 2 and Figure 3, Appendix A). These WD-unique haplotypes may possibly represent novel mutations occurring in weedy rice populations after their origin either from wild-introgression or de-domestication. In other words, these mutations are produced independently from wild and cultivated rice progenitors, although the likelihood of insufficient wild and cultivated rice samples being included in this study cannot be completely ruled out. Nevertheless, the possible novel mutations in weedy rice could serve as important sources for further evolution of weedy rice worldwide. This finding provides evidence for answering our third question in the Introduction, showing that weedy rice may further evolve by novel mutations after its origin to adapt to environmental changes. Further studies of the mutation-promoted weedy rice evolution linking with local environment conditions will help us to explore the underlying mechanisms of adaptive evolution in weedy rice.

Novel natural mutations can easily occur in agricultural weeds, including weedy rice populations that are exposed to diverse conditions in agroecosystems around the world with dramatic environmental changes [52,53]. All these findings suggest that novel mutation is very important for the new variation and adaptation of weedy rice populations infesting diverse rice-panting ecosystems in the world. Actually, a number of studies have already indicated WD-specific novel SNPs in weedy rice samples or populations [45,53], and some of these SNPs were identified to occur in the exon of the functional genes in weedy rice [27]. Consequently, novel mutations will give rise to the evolution of novel traits that are beneficial for weedy rice to adapt to changing environments.

De-domestication or mutation of cultivated crop species is a unique evolutionary process during which crops reacquire wild-like traits, such as seed shattering and seed dormancy, to survive and persist in human disturbed agroecosystems but without human assistance [21]. The reacquisition of seed dispersal functions is the crucial step for the origin and evolution of weedy rice de-domesticated from its crops. The main driving forces of evolution from domesticated types (crop) to weedy (wild-like) types include mutations and gene flow from wild and other genetically different weedy types. Weedy plants can possibly be derived from crop seeds that shattered before or during human harvesting. Plants that acquired wild-like traits through novel mutations, such as seed shattering, secondary dormancy, and modified phenotypes, have greater opportunities to be selected and adapt to the changing environment in local agroecosystems in which they can easily survive and reproduce without human intervention.

The ability to identify novel mutations or crop-weed gene flow is also important for us to understand the evolutionary processes of de-domesticated weedy rice populations. A recent study by Wang et al. [54] reported recurrent pollen-mediated gene flow that promotes genetic diversity and differentiation of weedy rice, indicating the important role of crop-weed introgression in weedy rice evolution. However, knowledge on how novel mutations influence the genetic diversity and differentiation of weedy rice is still limited. Therefore, weedy rice that evolved from the de-domestication pathway provides opportunities for scientists to explore the adaptive evolution of this conspecific weed generated by mutation-mediated de-domestication or crop-weed gene flow, the two important driving forces for evolution [55]. Revealing such evolutionary processes will also facilitate better understanding of the underlying mechanisms of the origin and evolution of other conspecific weeds infesting worldwide agricultural ecosystems.

## 5. Conclusions

As one of the most noxious weeds infesting rice fields worldwide, weedy rice poses great challenges for global food security by reducing the production of cultivated rice, a staple food consumed by one half of the world’s population. This conspecific weed has complex origins and abundant genetic diversity, which makes the control of weedy rice extremely difficult. We demonstrated in this study that most weedy rice had the de-domestication origins from cultivated progenitors and only a very small proportion of weedy rice had wild-introgression origins from its wild progenitor. This finding suggests that the de-domestication process, in which no wild progenitors were involved, plays the key role in the origins of weedy rice worldwide. In addition, weedy rice that evolved from de-domestication origins had a distinct genetic structure closely associated with the *indica* and *japonica* differentiation of cultivated rice. This finding evidently supports the proposed key role of de-domestication in the origins and evolution of weedy rice, and that weedy rice populations with de-domestication origins may have independent origins from *indica* or *japonica* cultivars. Furthermore, novel mutants as represented by the WD-unique haplotypes that were only identified in weedy rice suggest the continued adaptive evolution of weedy rice in different rice-planting regions after its origins. Such continued evolution of weedy rice adapting to different agricultural ecosystems has significantly promoted the genetic differentiation and diversity of weedy rice worldwide. The knowledge of weedy rice generated from this study provides a useful case study for deep insights into the origin and evolution of conspecific weeds, in addition to the design of effective measures to control these weeds.

## Figures and Tables

**Figure 1 biology-10-00828-f001:**
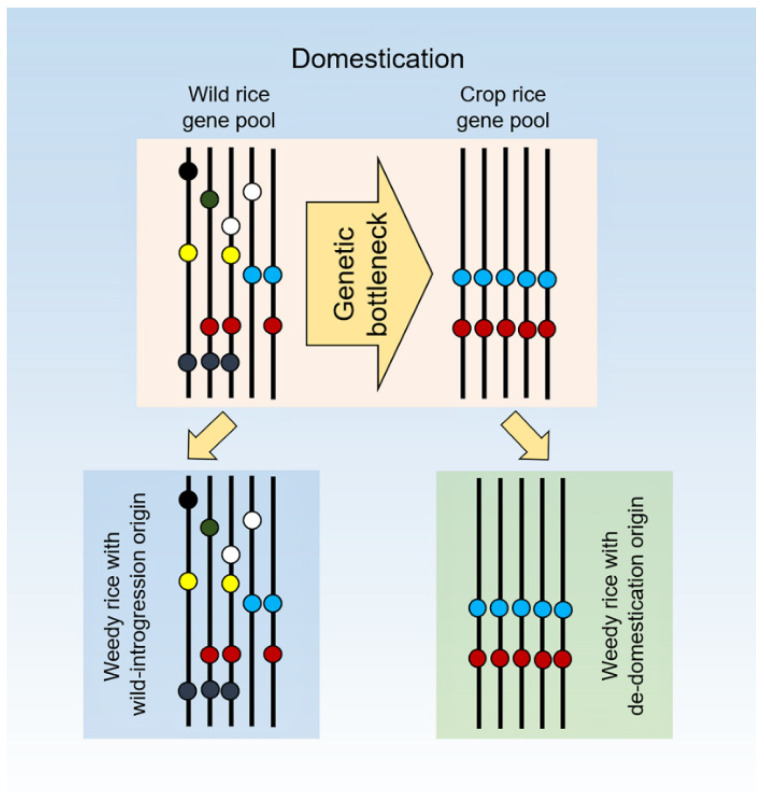
Hypothetical illustration indicating substantial reduction of single nucleotide polymorphisms (SNPs) at a DNA fragment of a candidate gene and its flanking regions from wild rice (top-panel, left) to crop rice (top-panel, right) gene pools, due to genetic bottlenecks in domestication. Consequently, weedy rice populations that evolved directly from wild rice (wild-introgression, bottom panel, left) should have a much greater level of SNPs than those that evolved from crop rice (de-domestication, bottom panel, right). Circles coded with colors represent different genotypes. Adapted from Ross-Ibarra et al. [30].

**Figure 2 biology-10-00828-f002:**
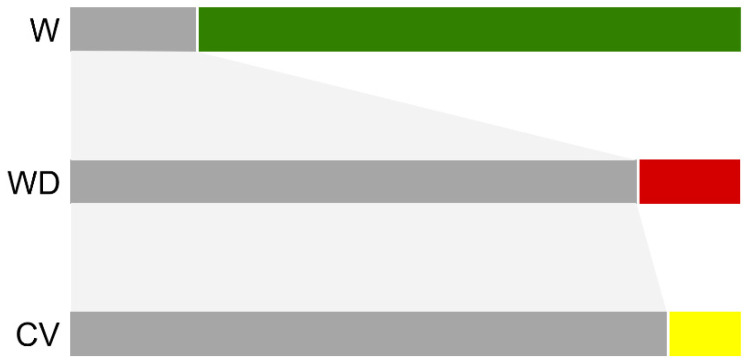
The proportion of different haplotypes of the *sh4* gene in wild (W), weedy (WD), and cultivated (CV) rice samples. The gray bars represent haplotypes shared by all three taxa. The green, red, or yellow bars represent the proportions of unique haplotypes in the examined wild, weedy, or cultivated rice samples.

**Figure 3 biology-10-00828-f003:**
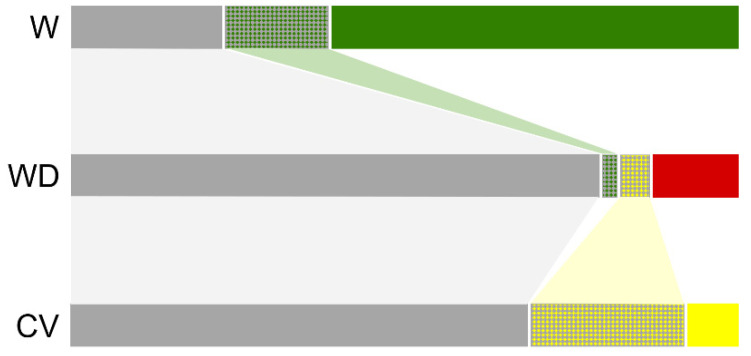
The proportion of different haplotypes of the *qsh1* gene in wild (W), weedy (WD), and cultivated (CV) rice samples. The gray bars represent the haplotypes shared by wild, weedy, and cultivated rice samples. The light-green and light-yellow bars indicate the proportion of specific haplotypes shared by wild and weedy or by weedy and cultivated rice samples, respectively. The green, red, or yellow bars represent the proportions of unique haplotypes in the examined wild, weedy, or cultivated rice samples.

**Figure 4 biology-10-00828-f004:**
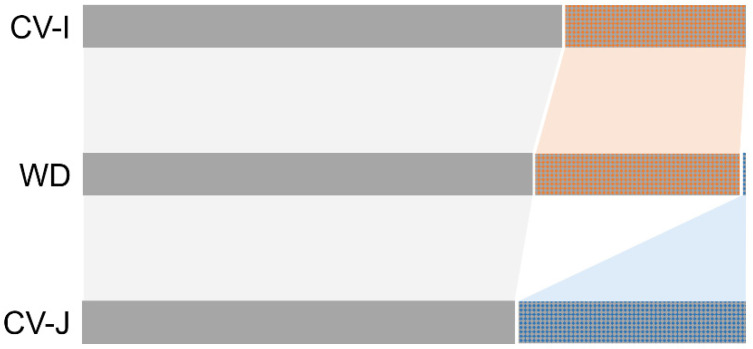
The proportion of different haplotypes of the *qsh1* gene shared by the examined weedy rice (WD) and cultivated rice (CV). The gray bars represent the haplotypes shared by the *indica* cultivated (CV-I), weedy (WD), and *japonica* cultivated (CV-J) rice samples. The light-brown and light-blue bars indicate the proportion of specific haplotypes shared by *indica* cultivated and weedy or by *japonica* cultivated and weedy rice samples, respectively.

**Figure 5 biology-10-00828-f005:**
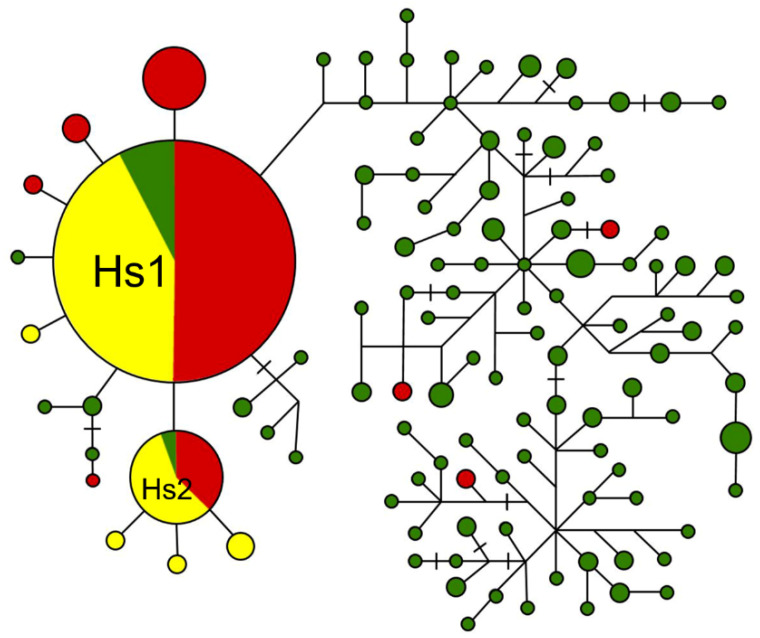
Haplotype networks constructed based on the DNA sequence polymorphisms of the *sh4* gene. The network shows WD samples clustered into two key groups with one (on the left) having a few haplotypes shared by most W (coded as red), CV (yellow), and a few W (green) samples, and another (on the right) having only three WD haplotypes nested among the W haplotypes. The size of the circles indicates the relative number of samples in the haplotypes. Lines between circles represent one-step variation. Dashes on the lines represent more than one step of variation/mutation.

**Figure 6 biology-10-00828-f006:**
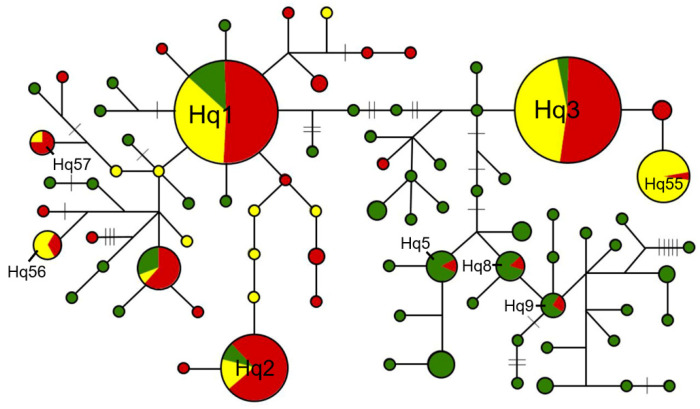
Haplotype networks constructed based on the DNA sequence polymorphisms of the *qsh1* gene. The network shows WD samples clustered into a few major groups with a few key haplotypes shared by WD (red), CV (yellow), and W (green) rice samples. Only a few WD haplotypes are nested among the W haplotypes. The size of the circles indicates the relative number of samples in the haplotypes. Lines between circles represent one-step variation. Dashes on the lines represent more than one step of variation/mutation.

**Figure 7 biology-10-00828-f007:**
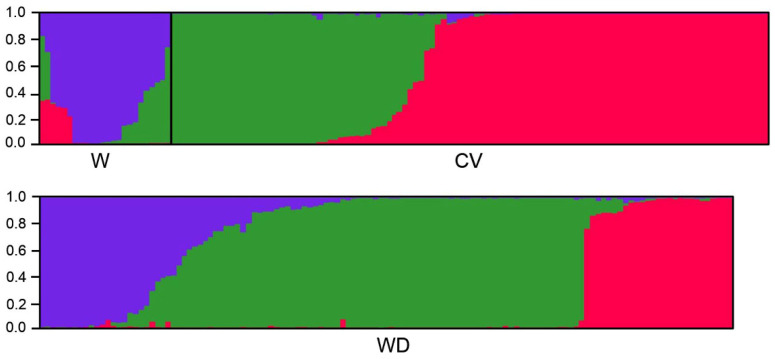
Genetic structure of wild (W), cultivated (CV), and weedy rice (WD) samples based on polymorphisms of the SSR (simple sequence repeat) fingerprints. The key genetic components of wild rice (blue), cultivated rice (green and red), and weedy rice (blue, green, and red) are identified. Each sample is represented by a single vertical line (bar) assigned into the three colored segments (genetic components), with lengths proportional to each of the three inferred clusters. Numbers at the vertical axis represent the probability of assignment.

**Figure 8 biology-10-00828-f008:**
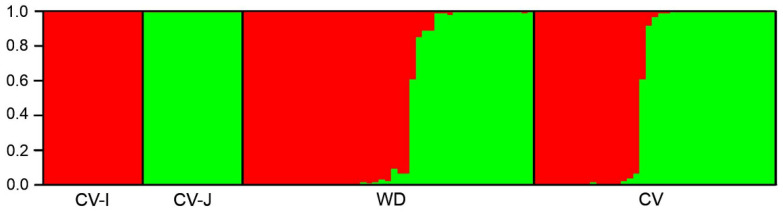
The genetic structure of standard *indica* (CV-I) and *japonica* (CV-J) varieties, weedy rice (WD), and cultivated rice (CV) samples based on the InDel (insertion/deletion) molecular fingerprints. When the K value was 2, CV-I (red) and CV-J (green) varieties showed two distinct genetic components. Accordingly, both WD and CV samples also showed the two distinct genetic components (red and green). Each sample is represented by a single vertical line (bar) assigned into the two-colored segments (genetic components), with lengths proportional to each of the two inferred clusters. Numbers at the vertical axis represent the probability of assignment.

**Table 1 biology-10-00828-t001:** The number of identified variable sites and haplotypes in the analyzed DNA sequences of the *sh4* and *qsh1* seed-shattering loci in wild, weedy, and cultivated rice samples.

Locus	Taxon	No. of Variable Sites	No. of Haplotypes	Proportion of Total Haplotypes ^1^	Type of Haplotypes	
					Shared Haplotypes	Specific Haplotypes ^2^
*sh4*	Wild rice	101	103	87.3%	Hs1, Hs2	Hs3~Hs103
	Weedy rice	35	9	7.6%	Hs1, Hs2	Hs104~Hs110
	Cultivated rice	4	6	5.1%	Hs1, Hs2	Hs111~Hs114
*qsh1*	Wild rice	70	54	57.4%	Hq1~Hq4	Hq5~Hq54
	Weedy rice	33	25	26.6%	Hq1~Hq4	Hq5, Hq8, Hq9, Hq55~Hq72
	Cultivated rice	24	15	16.0%	Hq1~Hq4	Hq55~Hq57, Hq73~Hq80

^1^ Based on the total number of 114 haplotypes in *sh4* locus and 80 haplotypes in *qsh1* locus., ^2^ Hq5, Hq8, and Hq9 were only shared by wild rice and weedy rice; Hq55~Hq57 were only shared by cultivated rice and weedy rice.

## Data Availability

All the data are shown in the article.

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
