# Peer review of "Key Roles of De-Domestication and Novel Mutation in Origin and Diversification of Global Weedy Rice"

_biology, 2021, doi:10.3390/biology10090828_

Round 1
Reviewer 1 Report
The authors' MS is a mature, well written text about the subject. It became understandable that just a few mutations are needed to transform a cultured rice variety to some that shatters seeds. Thus, the main conclusion (WD mostly seems to stem from CV) looks probable from that and it is nice to see the data supporting this preceding logical conclusion.
As reviewer, I enjoyed the good literature review/introduction to the topic, the 3 clear scientific questions and how the article tried to answer them. Though the methods are standard, their description is informative.
The results/discussion part is generally convincing though the reviewer feels that the authors try to avoid explicit quantitative conclusions. I was looking for a list of WD strains that originate (1) from CV or (2) from W or (3) where it is not clear to make a definitive conclusion. I would expect numbers like "A strains out of 169 WD are derived from CV and B varieties can be traced back to W. The remaining 169-A-B are in a twilight with regard to origin." Similarly, I would expect clear numbers with regard to the indica/japonica varieties that could accompany Figures 7 and 9. Generally, section 3.3 was written essentially without using numbers but only with provision of visual impressions.
Minor issues:
(1) line 178 "ClustalX" instead of "ClutalX"
(2) What is the case of Figure 8?
Author Response
POINT-BY-POINT RESPONSES TO THE REVIEWER’S COMMENTS
COMMENTS FROM REVIEWER #1
Comment 1. The authors' MS is a mature, well written text about the subject. It became understandable that just a few mutations are needed to transform a cultured rice variety to some that shatters seeds. Thus, the main conclusion (WD mostly seems to stem from CV) looks probable from that and it is nice to see the data supporting this preceding logical conclusion.
As reviewer, I enjoyed the good literature review/introduction to the topic, the 3 clear scientific questions and how the article tried to answer them. Though the methods are standard, their description is informative.
Response – We greatly appreciate the positive comments from the reviewer.
Comment 2. I would expect numbers like "A strains out of 169 WD are derived from CV and B varieties can be traced back to W. The remaining 169-A-B are in a twilight with regard to origin." Similarly, I would expect clear numbers with regard to the indica/japonica varieties that could accompany Figures 7 and 9. Generally, section 3.3 was written essentially without using numbers but only with provision of visual impressions.
Response – We are indeed thankful to the reviewer for the constructive suggestions, following which we calculated the proportion of weedy rice samples with different origins and have added the information into the revised manuscript (see the section 3.3 in Line 342-344 and Line 372-373). Roughly, ~20% of WD samples evolved from wild rice, while ~80% of WD samples from CV rice (Figures 5-7). About the comment of weedy rice associated with indica-japonica genetic differentiation, we indicated that ~58% WD samples with the same genetic component of indica CV rice, and ~42% samples of japonica CV rice (Figures 8), based on our calculation.
Comment 2.
Minor issues:
(1) line 178 "ClustalX" instead of "ClutalX"
Response – Following the reviewer’s comment, we have changed “ClutalX” to “CLUSTAL X” in Line 178 and in Line 156.
(2) What is the case of Figure 8?
Response – Thanks for the comment. We mistakenly labeled “Figure 8” as “Figure 9”. We have corrected the order of the Figures throughout the text.

Reviewer 2 Report
Authors analyzed DNA sequence polymorphisms of the seed shattering genes (sh4 and qsh1) in weedy, wild, and cultivated rice from a worldwide distribution. This approach is relatively new; methods are appropriate. The results are interesting and worth to be published, which demonstrate the key role of de-domestication in the origin of weedy rice and the evolution of conspecific weeds.
Author Response
POINT-BY-POINT RESPONSES TO THE REVIEWER’S COMMENTS
COMMENTS FROM REVIEWER #2
Comment 1. Authors analyzed DNA sequence polymorphisms of the seed shattering genes (sh4 and qsh1) in weedy, wild, and cultivated rice from a worldwide distribution. This approach is relatively new; methods are appropriate. The results are interesting and worth to be published, which demonstrate the key role of de-domestication in the origin of weedy rice and the evolution of conspecific weeds.
Response – We are deeply thankful to the positive comments from the reviewer.

Reviewer 3 Report
Very well organized, researched, developed and written manuscript. As an outsider to this specific set of research questions, I was able to follow the flow of your project and understood your results and conclusions. Strong contribution.
Author Response
POINT-BY-POINT RESPONSES TO THE REVIEWER’S COMMENTS
COMMENTS FROM REVIEWER #3
Comment 1. Very well organized, researched, developed and written manuscript. As an outsider to this specific set of research questions, I was able to follow the flow of your project and understood your results and conclusions. Strong contribution.
Response – We appreciate the reviewer’s support to the manuscript.
